# PRDM2—The Key Research Targets for the Development of Diseases in Various Systems

**DOI:** 10.3390/biom15081170

**Published:** 2025-08-15

**Authors:** Shiqi Deng, Hui Li, Chenyu Zhu, Lingli Zhang, Jun Zou

**Affiliations:** 1 College of Athletic Performance, Shanghai University of Sport, Shanghai 200438, China; 2421811003@sus.edu.cn; 2 School of Exercise and Health, Shanghai University of Sport, Shanghai 200438, China; lihui@suat-sz.edu.cn (H.L.); 2311517002@sus.edu.cn (C.Z.)

**Keywords:** biological function, diseases of various systems, PRDM2/RIZ, RIZ1, RIZ2, structural

## Abstract

PR/SET domain 2 (PRDM2)/RIZ is a member of the histone/protein methyltransferases (PRDMs) superfamily. Discovered to have the ability to bind retinoblastoma in the mid-1990s, PRDM2 was assumed to play a role in neuronal development. Like other family members characterized by a conserved N-terminal PR structural domain and a classical C2H2 zinc-finger array at the C-terminus, PRDM2 encodes two major protein types, the RIZ1 and RIZ2 isoforms. The two subtypes differ in the presence or absence of the PR domain: the RIZ1 subtype has the PR domain, whereas the RIZ2 subtype lacks it. The PR domain exhibits varying conservation levels across species and shares structural and functional similarities with the catalytic SET domain, defining histone methyltransferases. Functioning as an SET domain, the PR domain possesses protein-binding interfaces and acts as a lysine methyltransferase. The variable number of classic C2H2 zinc fingers at the C-terminus may mediate protein–protein, protein–RNA, or protein–DNA interactions. An imbalance in the RIZ1/RIZ2 mechanism may be an essential cause of malignant tumors, where PR-positive isoforms are usually lost or downregulated. Conversely, PR-negative isoforms are always present at higher levels in cancer cells. RIZ1 isoforms are also important targets for estradiol interaction with hormone receptors. PRDM2 can regulate gene transcription and expression combined with transcription factors and plays a role in the development of several systemic diseases through mRNA expression deletion, code-shift mutation, chromosomal deletion, and missense mutation occurrence. Thus, PRDM2 is a key indicator for disease diagnosis, but it lacks systematic summaries to serve as a reference for study. Therefore, this paper describes the structure and biological function of PRDM2 from the perspective of its role in various systemic diseases. It also organizes and categorizes its latest research progress to provide a systematic theoretical basis for a more in-depth investigation of the molecular mechanism of PRDM2’s involvement in disease progression and clinical practice.

## 1. Introduction

PRDMs are a subfamily of Kruppel-like zinc finger (ZF) gene proteins, first appearing in holozoans—a group comprising metazoans and their closest unicellular relatives—and subsequently expanding across vertebrates before undergoing further duplications in primates [1]. PR structural domain-containing 2 (PRDM2) is in the chromosomal region (1p36), and this region is commonly influenced by genetic alterations in a variety of malignant tumors. It exerts a tumor-suppressor effect and plays an essential role in regulating animal development and cancer progression [2]. Meanwhile, PRDM2 is one of the classical genes of the PRDM family with a PR structural domain at the N-terminal end and a classical C2H2 ZF toward the C-terminal end. It functions as a transcription factor in the nucleus and encodes two significant proteins. They are RIZ1 at 280 kDa, which contains the PR structural domain, and RIZ2 at 250 kDa, which is deficient in the PR structural domain. RIZ1 and RIZ2 are extensively expressed in similar ratios in normal tissues but have opposite functions in cancer, and their imbalanced ratios are also assumed to be a factor contributing to the pathogenesis of malignant tumors [3]. Accordingly, the mechanism of RIZ1/RIZ2 balance may be involved in developing various systemic diseases by regulating the multiplication of many cells and thus in disease processes. Therefore, our review studies PRDM2 in various systemic diseases and discusses the significant advances in its physiological role and mechanism of action.

## 2. Structure and Biological Functions of PRDM2

### 2.1. Structure of PRDM2

PRDM2 is a member of the PRDM family and exhibits the characteristic structural features of this protein group. It was initially identified as a retinoblastoma (Rb)-interacting zinc-finger protein and is thus also referred to as a retinoblastoma-interacting zinc finger (RIZ) protein [4,5].The structure of PRDM2 is shown in Figure 1. 

**Figure 1 biomolecules-15-01170-f001:**
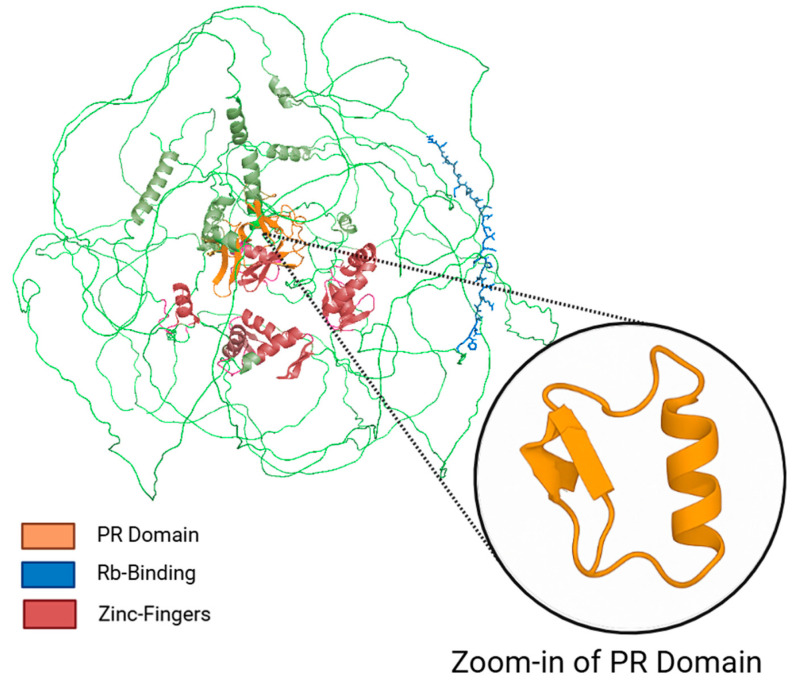
Structural representation of the PRDM2 protein highlighting functional domains. The three-dimensional structure of PRDM2 was predicted using AlphaFold2. Key functional domains are color-coded: the PR (PRDI-BF1 and RIZ) domain is shown in orange, Rb-binding motifs in blue, and C2H2 zinc finger motifs in red. The protein backbone is rendered in green. The inset provides a zoom-in view of the PR domain, highlighting the structural features underlying its functional importance. The spatial arrangement of these domains suggests potential roles in transcriptional regulation and chromatin interaction. Note: The structural model was obtained from the AlphaFold Protein Structure Database (https://alphafold.ebi.ac.uk/entry/Q13029 (accessed on 14 August 2025)) and visualized using PyMOL 3.1 (Schrödinger, LLC, New York, NY, USA). Domain annotation was performed manually based on UniProt 2025_03 and the literature references.

Among its several isoforms, the two most studied are PRDM isoform1 (RIZ1, UniProt ID: Q13029-1) and PRDM2 isoform2 (RIZ 2, UniProt ID: Q13029-3), which differ primarily in the presence or absence of the N-terminal PR domain. Afterwards, a PRDM2 variant, named MTE-binding protein ZF type (MTB-Zf, UniProt ID: Q13029-2), has been isolated from a human monocytic leukemia cell line cDNA expression library. This variant is downregulated by binding to the cis-regulatory DNA element GTCATTGAC (MTE: macrophage-specific TPA-responsive element), which is responsible for the induction of the human heme oxygenase-1 gene during macrophage differentiation [6]. The major PRDM2 isoforms and their key structural and functional characteristics are summarized in Table 1.

**Table 1 biomolecules-15-01170-t001:** Comparison of major PRDM2 isoforms: structural characteristics and functional features.

Isoform Name	UniProt ID	Structural Domains	Key Features	Notes
RIZ1 (PRDM2 Isoform 1)	Q13029-1	PR domain (N-terminal), Rb-binding motif, 8 C2H2 zinc fingers	Full-length isoform; exhibits H3K9 methyltransferase activity; functions as tumor suppressor	Nuclear localization; involved in chromatin remodeling and cell cycle regulation
RIZ2 (PRDM2 Isoform 3)	Q13029-3	Rb-binding motif, 8 C2H2 zinc fingers (lacks PR domain)	Shorter isoform via alternative promoter; lacks methyltransferase activity	May antagonize RIZ1; often upregulated in cancers
Transcript 3/RIZ2 isoform c	Not assigned	Similar to RIZ2 (lacks PR domain)	Variant transcript regulated by estrogen receptor α in monocytic leukemia cells	Functional role remains unclear; may overlap with RIZ2 but differs in hormonal responsiveness
MTB-Zf (PRDM2 Isoform 2)	Q13029-2	8 C2H2 zinc fingers (lacks PR domain)	Variant identified from monocytic leukemia cells; MTE-responsive	Limited functional data; classified as PRDM2 isoform 2 in UniProt

Note: Transcript 3/RIZ2 isoform c is reported in specific studies as an estrogen-responsive variant lacking the PR domain. Its distinction from canonical RIZ2 remains to be clarified at the protein level, and no independent UniProt entry is currently available.

However, the original study did not clarify the mechanistic link between MTE binding and the downregulation of MTB-Zf expression; the observed decrease may be secondary to broader transcriptional changes during macrophage differentiation.

The full-length RIZ1 transcript contains 10 exons and encodes a nuclear phosphoprotein of 1718 amino acids (188.9 kDa). RIZ1 is highly conserved among species, showing ~84% sequence similarity to the mouse and rat homologs. It possesses histone methyltransferase activity via its PR domain, specifically catalyzing H3K9 methylation, thereby participating in chromatin remodeling and transcriptional repression [7,8]. This epigenetic silencing is associated with the inhibition of cell proliferation [9]. RIZ1 can also interact with Rb proteins both in vitro and in vivo, further supporting its regulatory role in cell cycle control.

An additional isoform, sometimes referred to as transcript 3 or RIZ2 isoform c, lacks the PR domain and shows estrogen responsiveness mediated by estrogen receptor alpha (ERα) [9]. Despite its identification, RIZ2 isoform c has not been extensively characterized, and further studies are required to elucidate its biological significance.

Notably, both RIZ1 and RIZ2 are involved in estrogen signaling, although in different ways. RIZ2 is shorter due to the absence of the PR domain, and while neither isoform possesses a strong intrinsic repressor function, RIZ1 often acts as a more potent repressor in specific promoter contexts—suggesting that the PR domain may modulate transcription indirectly [10,11].

In distribution, the two isoforms RIZ1 and RIZ2 have a generalized nuclear localization. Although a complete picture of the developmental or tissue distribution of the different isoforms is not yet available, published data describe comparable ratios (1:1) of RIZ1 to RIZ2 transcripts in most tissues, such as the brain, heart, skeletal muscle, kidney, liver, and spleen [10]. The expression RIZ1 and RIZ2 in similar ratios induces significant changes in the expression of key transcription factors involved in T-lymphocyte differentiation, and imbalanced ratios are a factor affecting the pathogenesis of malignant tumors [8,12]. However, the level of RIZ1 transcripts is 5- to 10-fold higher in the testis than in RIZ2 [10]. RIZ1 inactivation is usually found in many types of human cancers, occurring through loss or silencing of RIZ mRNA expression, frameshift mutations, chromosomal deletions, and missense mutations. Specifically, targeted deletion of the RIZ1 isoform (but not RIZ2) renders mice significantly more susceptible to tumor formation [13,14]. The PR-Set7 binding domain, located within the N-terminal PR domain of RIZ1, has been shown to interact with histone methyltransferases and contribute to chromatin-based transcriptional repression. This domain plays a critical role in mediating the tumor-suppressive activity of RIZ1 by facilitating nuclear localization and the establishment of histone modification patterns (e.g., H4K20me1-H3K9me1) essential for the repression of oncogenic targets [13,14,15]. On the other hand, the methylation of the RIZ1 promoter is closely associated with the loss or reduction in RIZ1 mRNA expression in breast, hepatocellular, colon, and lung cancer cell lines, as well as in hepatocellular carcinoma tissues. Additionally, DNA methylation is a common mechanism for inactivating the RIZ1 oncogene in human hepatocellular carcinoma and breast cancer [16]. While RIZ1 is generally considered a tumor suppressor, some findings may reflect the mixed or ambiguous effects of PRDM2 isoforms in specific contexts. Therefore, caution is required when interpreting oncogenic roles in studies where isoforms were not clearly distinguished.

Therefore, the structure defines its function, and PRDM2 is widely present in various organisms with a unique structure that determines its biological function in the progression of diseases in various systems of the organism. This paper summarizes and generalizes this to provide a systematic theoretical basis for subsequent research and clinical practice.

### 2.2. Biological Function of PRDM2

#### 2.2.1. The Role of PRDM2 in Cellular Metabolic Processes

PRDM2/RIZ shows strong upstream–downstream dependence by selecting different target promoters and binding sites, conferring the ability to drive cell differentiation and specify cell fate choices. Additionally, PRDM2 participates in the signal transduction of many cells. An overview of PRDM2’s roles in different cellular metabolic and physiological processes is shown in Figure 2.

RIZ proteins can regulate myeloid differentiation, which is involved in developing myelodysplastic syndromes [17,18]. Research shows that a high expression of the RIZ1 protein in human multiple myeloma cells inhibits cell proliferation and induces caspase-dependent apoptosis [19]. In chronic granulocytic leukemia, matricellular transformation is correlated with the deletion of heterozygosity in the region where RIZ1 is, as well as with reduced RIZ1 expression and overexpression of RIZ1 in a chronic granulocytic leukemia model cell line that inhibits cell proliferation, exacerbates apoptosis, and enhances cell differentiation [20]. Conversely, the reduction in RIZ1 activity leads to decreased apoptosis and differentiation and enhanced proliferation, which in turn leads to an increased number of myeloblasts [21]. Figure 2The role of PRDM2 in cellular metabolic and physiological processes. This figure illustrates the diverse biological functions of PRDM2 across multiple systems. In muscle tissue, PRDM2 maintains quiescence by regulating H3K9 methylation, and its knockdown leads to loss of muscle quiescence. In bone, PRDM2 promotes osteoclast differentiation by upregulating NFATc1 and TRAP expression. In the immune system, PRDM2 (RIZ1 isoform) modulates NF-κB signaling and promotes the expression of inflammatory cytokines IL-6 and TNF-α. In hematopoietic cells, PRDM2 suppresses proliferation through RIZ1-mediated mechanisms. In metabolic regulation, PRDM2 influences obesity-related pathways via AKT3 signaling and H3K9 methylation. Collectively, these findings highlight PRDM2’s pivotal role in transcriptional regulation and cellular homeostasis.
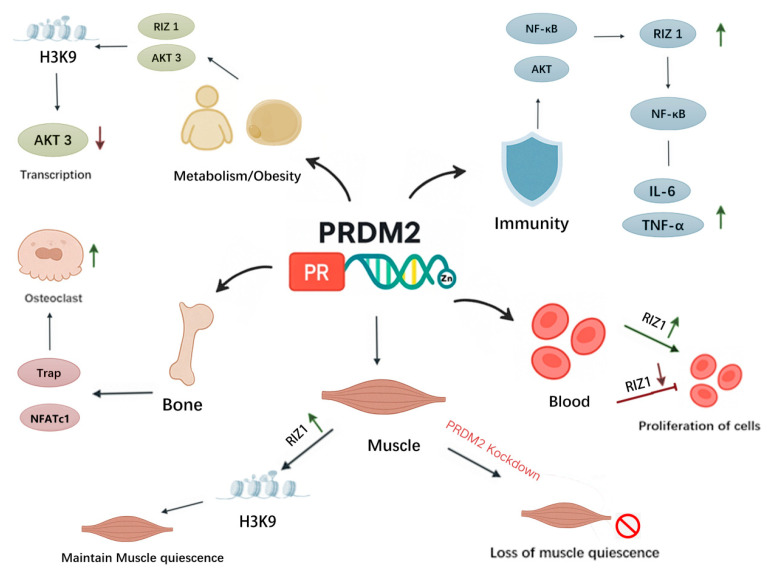


The dormancy of adult stem cells is essential to ensuring regeneration and reducing tumorigenesis. Epigenetic regulation contributes to cell-cycle control and differentiation. PRDM2/RIZ is abundant in quiescent muscle stem cells and controls the reversible quiescence of cultured adult myoblasts. PRDM2/RIZ knockdown alters the histone methylation of key promoters and disrupts the silencing program by the overall de-inhibition of myogenesis and the super-inhibition of the cell cycle. These results suggest that the epigenetic regulation of PRDM2/RIZ preserves the critical function of inactivity, which may be related to stem cell self-renewal [22]. However, the RNA interference and chromatin immunoprecipitation assays employed in this study do not discriminate between the two PRDM2 isoforms, because the shRNA and the antibody both target sequences common to RIZ1 and RIZ2. Consequently, the functions ascribed to PRDM2 herein should be regarded as the combined or synergistic outcome of both RIZ1 and RIZ2; the specific contribution of either isoform remains to be determined.

Through epigenetic regulation, especially the role of PRDM2/RIZ in quiescent muscle stem cells, the dormancy of adult stem cells ensures a balance between cell-cycle control and differentiation. Moreover, RIZ1 acts downstream of different signaling pathways to promote cell maturation and differentiation. A study has found that lipopolysaccharide stimulated RIZ1 expression in RAW 264.7 macrophage-like cells by activating the NF-κB and AKT signaling pathways, and RIZ1 can synergize with p53 to enhance the lipopolysaccharide-induced activation of the NF-κB signaling pathway, accompanied by pro-inflammatory cytokine production [23]. Moreover, mouse RAW 264.7 macrophage-like cells RIZ1 mediate the osteoclast formation induced by the NF-κB ligand–receptor activator RANKL. The expression of RIZ1 was significantly enhanced following RANKL treatment. Moreover, silencing of RIZ1 significantly inhibited the expression of TRAP and NFATc1 [24]. These findings suggest that RIZ1 may participate in RANKL-induced osteoclastogenesis by regulating the expression of NFATc1. RIZ1 is an essential regulator of AKT3 transcription and AKT phosphorylation, suggesting a role for RIZ1 in high-fat diet-induced obesity and the AKT pathway [25]. RIZ1 overexpression in HEK293 cells decreases the expression of the AKT3 protein and significantly reduces the luciferase reporter activity of the AKT3 gene promoter in cells co-transfected with RIZ1. The RIZ1 recombinant protein can further bind the AKT3 promoter in vitro, and chromatin immunoprecipitation assays also demonstrate that RIZ1 can bind the AKT3 promoter in vivo. RIZ1 overexpression increases H3K9 methylation on the AKT3 promoter as well. All these results indicate that AKT3 is a target of RIZ1 regulation, thereby expanding our understanding of the AKT pathway in cancer and obesity [26].

#### 2.2.2. PRDM2 and Estrogen Receptor-Specific Binding

RIZ1 functions as a specific coactivator of estrogen receptor-mediated transcription and acts as a downstream effector of estrogen signaling in target tissues. Estradiol binding converts RIZ1 activity from a transcriptional repressor to an activator in a ligand-dependent manner, implicating it in the regulation of organ-specific hormonal responses. Its absence in RIZ1-deficient but RIZ2-expressing murine models leads to impaired estrogen responses, underscoring its essential and non-redundant role in hormone-dependent gene regulation [26,27]. After estradiol induction, RIZ1 mRNA expression selectively decreases but the total RIZ mRNA expression increases in MCF-7 cells. Silencing RIZ1 expression stimulates the proliferation of MCF-7 cells, an effect consistent with the action of estradiol [28]. Interestingly, the MCF-7 cell line, which expresses a fusion protein of the RIZ protein ZF protein, grows at a higher rate than either the parental or control cell lines, whether under hormone deprivation conditions or estrogen stimulation, conclusively demonstrating that the ZF structural domain possesses the oncogenic activity of the RIZ2 gene product [29]. RIZ1 expression is downregulated in prostate cancer cells, and its subcellular localization shifts from the nucleus to the cytoplasm as the cancer level progresses. In vitro, dihydrotestosterone or estradiol can induce high expression of RIZ1 in prostate epithelial cells, suggesting a possible role for estrogen receptor-mediated pathways in non-classical estradiol-targeted tissues [30]. RIZ silencing also reduces cell proliferation and increases apoptosis in both estrogen-responsive cells, MCF-7, and non-responsive cells, MDA-MB-231. This finding complements new insights into the putative tumor-promoting function of RIZ2 [31].

Additionally, estrogen combined with PRDM2 plays a role in osteoporosis and Parkinson’s disease. Decreased bone mineral density is a major risk factor for osteoporotic fracture, in which estrogen plays an important role in maintaining bone mass by binding to ER-α, and RIZ1 is a specific ER-α coactivator that strongly enhances its function in vivo and in vitro. Thus, RIZ1 may be a chemotherapeutic target for osteoporosis [32,33]. At all ages, the prevalence of Parkinson’s disease is much higher in men than in women, and these differences may be a consequence of the neuroprotective effects of estrogen on the nigrostriatal pathway. The relation of common variants in four estrogen-related genes with Parkinson’s disease has been explored. The results confirm the association of the PRDM2 variant with susceptibility to Parkinson’s disease, which is particularly prevalent in women [34].

## 3. PRDM2 Is the Key Research Target for Disease Development in Various Systems

PRDM2, particularly its RIZ1 subtype, serves as a crucial histone methyltransferase that plays a significant role in regulating cell proliferation, differentiation, apoptosis, and the maintenance of stem cell quiescence. In normal tissues, the RIZ1 and RIZ2 subtypes maintain a relatively stable expression ratio. However, in various malignant tumors, this ratio is severely disrupted, with a significant downregulation or functional inactivation of RIZ1 expression, indicating its core position as a tumor suppressor gene. PRDM2 participates in the occurrence and development of various diseases, including those in the digestive, nervous, reproductive, and endocrine systems, through multiple mechanisms such as epigenetic mechanisms (such as promoter hypermethylation), interaction with transcription factors, and signal pathway regulation (such as AKT and NF-κB), suggesting its potential as a novel therapeutic target for multiple systemic diseases. Figure 3 summarizes the roles of PRDM2 in various systemic diseases.

**Figure 3 biomolecules-15-01170-f003:**
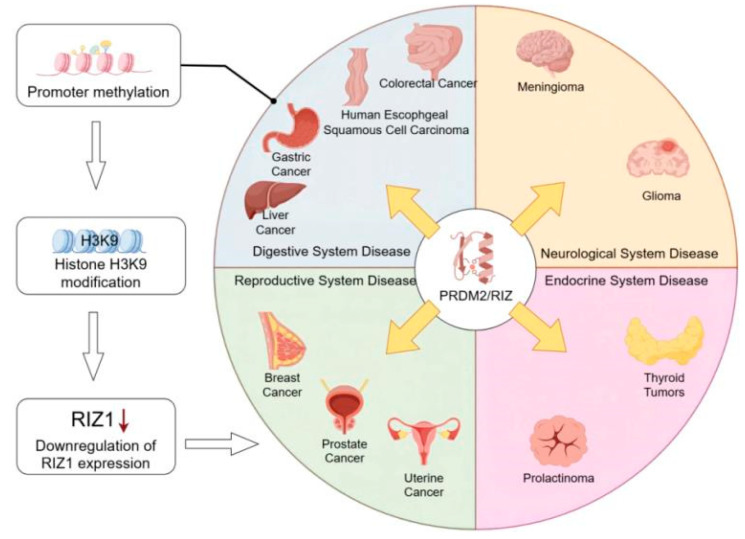
Mechanisms of PRDM2 action in diseases across different systems. The figure illustrates the role of the PRDM2 gene and its mechanisms in diseases across different systems. Promoter methylation affects RIZ1 expression, and its inactivation or mutation is closely associated with the development of various diseases, particularly in the development and progression of tumors, thereby promoting carcinogenesis.

### 3.1. Role of PRDM2 in Digestive Diseases

PRDM2 plays a role in multiple organ diseases of the digestive system, including the liver, esophagus, and stomach.

#### 3.1.1. PRDM2 and Liver Cancer

The liver is the largest digestive gland of the digestive system with functions in detoxification, metabolism, bile secretion, and immune defense. Liver dysfunction affects the health of the whole organism, for example, liver cancer. The distal short arm of human chromosome 1 (1p36) is usually altered in primary hepatocellular carcinomas and cell lines, whereas the PRDM2 gene is in this region. The expression of RIZ1 is reduced in hepatocellular carcinoma tumors, but RIZ2 is essentially unchanged. RIZ1 overexpression also inhibits the tumorigenicity of hepatocellular carcinoma cells in nude mice. Thus, the concept of the yin and yang of the RIZ gene product in human cancers is suggested, thereby also demonstrating that RIZ1 has an inhibitory effect on hepatocellular carcinoma [35]. The induction of RIZI promoter demethylation in hepatocellular carcinoma exacerbates tumor cell apoptosis and inhibits cell proliferation and migration [36]. The epigenetic inactivation of RIZ1 in hepatocellular carcinoma involves DNA methylation and histone modification, and promoter methylation and H3K9 modification work together to silence RIZ1. A cooperative effect subsequently ensues between these epigenetic modifications [37]. The characteristic pattern of DNA methylation alterations can be used to predict human hepatocarcinogenesis, in which PRDM2 shows a clear analogous clustering [38]. However, a study reported that an alternatively spliced variant of RIZ1 (RIZ1v2) is upregulated in hepatocellular carcinoma (HCC) cells, promoting tumor progression through an HOTAIRM1/miR-125b axis [39]. HOTAIRM1 acts as a ceRNA, sponging the tumor-suppressive miR-125b, which relieves repression on RIZ1v2 and enhances HCC cell proliferation and metastasis. These findings suggest isoform-specific roles of PRDM2 and highlight the importance of distinguishing RIZ1 variants in cancer research. Additionally, the hepatitis B virus induces hepatocellular carcinoma primarily through epigenetic alterations, which are accompanied by the inactivation of RIZ1 [40]. Therefore, PRDM2 and its isoforms may serve as potential biomarkers for the diagnosis and treatment of hepatocellular carcinoma.

#### 3.1.2. PRDM2 and Esophageal Squamous Carcinoma

The esophagus is a muscular tube that connects the pharynx to the stomach, facilitating the transit of food and liquids. Its inner lining consists of stratified squamous epithelium, which is highly susceptible to carcinogenic insults such as chronic thermal injury, alcohol consumption, and tobacco exposure. This intrinsic vulnerability contributes to the development of esophageal squamous cell carcinoma (ESCC), a malignancy with substantial global incidence and marked geographic variation [41]. The methylation status of the RIZ1 promoter region in human esophageal squamous carcinoma cell lines has been examined to verify the relationship of RIZ1 methylation with the occurrence, progression, and metastasis of human esophageal squamous carcinoma. The results show that the methylation rate of cancer tissues significantly increases, suggesting that promoter methylation may play an important role in the epigenetic silencing of RIZ1 expression in human esophageal squamous carcinomas. Moreover, RIZ1 is further recognized as a biomarker for detecting early-stage human esophageal squamous carcinomas [42]. RIZ1 is also recognized as a biomarker for the detection of early human esophageal squamous carcinoma. Gene-expression profiles change as well after transfection with RIZ1. Most differentially expressed genes are associated with cellular development, supervision of viral replication, co-stimulation of lymphocytes, and the development of the immune system in esophageal cells, suggesting that esophageal cancer development is associated with RIZ1 inactivation [43]. SMYD3 inhibition has further been noted to promote RIZ1 expression in TE13 cells, suggesting that the SMYD3-RIZ1 signaling pathway may be a therapeutic target for esophageal squamous carcinoma [44].

#### 3.1.3. PRDM2 and Gastric Cancer

The stomach, a pivotal organ proximally situated within the gastrointestinal tract, serves dual roles in food storage and initiating enzymatic digestion. Its glandular epithelium, chronically exposed to exogenous carcinogens (e.g., dietary nitrosamines, Helicobacter pylori infection, and bile reflux), exhibits marked susceptibility to adenomatous hyperplasia and neoplastic progression due to cumulative genomic instability and epigenetic dysregulation [45]. The double allele inactivation of the RIZ1 gene in human gastric cancer cells suggests that RIZ1 is also a specific target for human gastric carcinogenesis [46]. Researchers found the hypermethylation of the RIZ1 promoter in gastric cancer tissues, accompanied by a decrease in RIZ1 expression. In gastric cancer cell lines, the RIZ1 promoter is hypermethylated and RIZ1 transcription inactivation is observed, and treatment with a democratizing agent restores the transcription level of RIZ1. The above results reconfirm that the transcription inactivation of the RIZ1 gene may be associated with gastric cancer development [47]. Moreover, colorectal cancer (CRC) is one of the most common causes of cancer-related deaths worldwide. CRC is accompanied by the genetic inactivation of RIZ, suggesting that RIZ may be one of the targets for CRC diagnosis and gene therapy [48,49]. In addition to RIZ1, the role of RIZ2 in human CRC has become important and has been investigated accordingly [50]. The PRDM2 gene is frequently mutated and transcriptionally deregulated in CRC, and an increase in RIZ2 is highly correlated with a significant downregulation of RIZ1. The overexpression of RIZ2 induces profound changes in the transcriptome of CRC cells through the dysregulation of the EGF pathway, suggesting RIZ2 involvement in the autocrine regulation of the cell behavior of DLD1 cells by EGF. The autocrine regulation of the cell behavior of DLD1 cells is also a key factor affecting RIZ2 gene development. RIZ2 overexpression induces profound changes in the transcriptome of CRC cells through EGF-pathway dysregulation, suggesting that RIZ2 is involved in the autocrine regulation of DLD1 cell behavior by EGF. Notably, enforced RIZ2 expression increases cell viability, growth, colony formation, migration, and organoid formation. These effects may be mediated by the release of high levels of EGF from DLD1 cells overexpressing RIZ2 [51].

In conclusion, promoter hypermethylation of the PRDM2/RIZ gene leads to transcriptional silencing of the tumor-suppressive isoform RIZ1, thereby altering its biological functions and contributing to carcinogenesis in major digestive system organs.

### 3.2. Role of PRDM2 in Neurological Diseases

The nervous system is composed of nerve cells (neurons) and glia, and their lesions are accompanied by functional changes in PRDM2/RIZ. The development of pheochromocytoma and abdominal paraganglioma is closely associated with the recurrent inactivation of RIZ1 [52]. Low expression of RIZ1 may inhibit neuroblastoma development, but RIZ1 promoter methylation is uncommon in neuroblastoma, and its mechanism needs clarification [53]. RIZ1 overexpression in malignant meningioma cell lines inhibits tumor-cell proliferation, induces apoptosis, and blocks tumor cells in the G2/M phase of the cell cycle. Moreover, meningioma progression is accompanied by a significant downregulation of RIZ1 expression, suggesting that RIZ1 may be a target gene for malignant meningioma therapy [54]. High RIZ1 expression indicates a low glioma grade and is associated with delayed tumor progression and overall survival, suggesting that RIZ1 may be a more promising therapeutic target for gliomas [55]. RIZ1 is downregulated in high-grade meningiomas, and RIZ1 overexpression inhibits the proliferation and promotes the apoptosis of the IOMM–Lee malignant meningioma cell line. The n-terminal PR structural domain of RIZ1 is found to have growth-inhibitory and anticancer activities in primary human meningioma cells [56]. Clinicopathological evaluation also suggests that RIZ1 hypermethylation is negatively correlated with tumor grade and patient age, and the RIZ1 promoter is hypermethylated in human glioblastoma cell lines. This finding suggests that promoting hypermethylation may play an important role in the epigenetic silencing of RIZ1 expression in human glioma tissues and glioblastoma cell lines [57]. Studies have shown that RIZ1 expression is correlated with tumor progression and treatment response, indicating its potential as a potential therapeutic target and suggesting that the PRDM2 gene further influences brain–tumor development by regulating the proliferation and differentiation of neural cells.

In summary, the hypermethylation of the PRDM2/RIZ gene promoter is also one of the pathogenic mechanisms of neurological diseases, such as meningiomas and gliomas, suggesting that PRDM2/RIZ may be one of the targets for the diagnosis and prevention of neurological diseases. The nervous system directly or indirectly regulates the functions of various organs and systems in the human body, thereby playing a dominant role in an organism. Therefore, PRDM2/RIZ may have great research significance in cross-organ and system regulation.

### 3.3. Role of PRDM2 in Reproductive Disorders

The 1p36 region where RIZ is located is believed to be a host for breast cancer suppressor genes. A study finds that RIZ1 expression is reduced or undetectable in breast cancer tissues and cell lines, with no significant change in RIZ2. Additionally, RIZ1 overexpression in breast cancer cells reportedly results in G2-M cell-cycle arrest and/or programmed cell death. These observations suggest an exclusive negative selection for RIZ1, but not for RIZ2, in breast cancer [58]. Abnormal DNA methylation induces RIZ1 transcriptional inactivation during prostate carcinogenesis, suggesting that RIZ gene alterations may be associated with epigenetic changes in prostate cancer [59]. Studies suggest that PRDM2 also plays a potential tumor-suppressor role in testicular germ–cell tumor formation [60]. RIZ1 gene therapy inhibits the growth of human cervical squamous carcinoma cells and synergistically enhances the therapeutic efficacy with clinical medications [61]. Whole-genome sequencing of blood and cancerous tissues from patients with malignant mesothelioma of the vaginal testicular membrane reveals mutations in the PRDM2 gene, suggesting that PRDM2 may serve as a target for the development of malignant mesothelioma of the vaginal testicular membrane [62]. Overall, RIZ genes may be therapeutic targets in various cancers, showing their importance in tumor biology. It also proves once again that structure determines function, and that the two isoforms RIZ1 and RIZ2 play different roles depending on the presence or absence of the PR structural domains. Moreover, the balancing mechanism of RIZ1/RIZ2 in the reproductive system warrants further investigation.

The above findings suggest that PRDM2 mediates cancers of breast, prostate, and other reproductive organs, and can be one of the gene targets for disease prevention, diagnosis, and treatment, and has been proven. However, it is presently limited to sequencing screening, and protein–protein interactions and drug–protein relationships have not yet been clearly elucidated and need further validation.

### 3.4. Role of PRDM2 in Endocrine System Diseases

Alterations of the RIZ1 locus in parathyroid tumors are found to act through intragenic allelic deletion and promoter hypermethylation [63]. RIZ1 expression is absent in thyroid tumor cell lines and is significantly reduced in thyroid cancer tissues. The aberrant cytosine methylation of the RIZ1 promoter mediates RIZ1 deletion, which can be reversed by intervention with DNA methyltransferase inhibitors. The above results suggest that RIZ1 plays an important role in thyroid tumorigenesis and is a potential new therapeutic target [64]. Prolactinoma is a common disorder in hypothalamic–pituitary diseases caused by excessive prolactin secretion from pituitary prolactin cell tumors. It occupies the first place in the incidence of functional pituitary tumors, with a higher incidence in women than in men. Reduced PRDM2 expression is associated with dopamine–agonist resistance and tumor recurrence in prolactinomas [65].

Endocrine glands and tissue cells secrete hormones that are released directly into the blood or lymphatic fluid, circulate through the bloodstream, and are transported throughout the body to act on certain target organs or target cells that can be acted upon, thereby regulating their physiological activities. Among them, PRDM2 mediates the progression of parathyroid tumors and lactinomas, but there are fewer related studies with their limitations to be added.

Additionally, promoter hypermethylation of the RIZ1 gene is prevalent in nasopharyngeal carcinoma, thereby playing a key role in clinical applications such as nasopharyngeal carcinoma screening and healing monitoring [66]. The inactivation of RIZ in cancer primarily includes code-shift mutations, hypermethylation, and missense mutations, among which RIZ polymorphisms may be important predictive markers of lung cancer susceptibility [67]. RIZ1 also has a regulatory role in human osteosarcoma cells and tissues [68]. T-cell prolymphoblastic leukemia is an aggressive malignancy in which epigenetically regulated PRDM2 genes are recurrently mutated [69]. RIZ is recognized as a tumor suppressor candidate gene on 1p36, and genetic alterations are also seen in this region of malignant melanoma. Transcoding mutations in the RIZ gene are found in melanoma and nevus samples, thereby suggesting a potential role for RIZ in the multistep process of tumor formation in cutaneous malignant melanoma [70]. The PRDM2 protein acts upon T-lymphocyte activation, whereas the mechanism and function of PRDM2 in lymphocyte activation/differentiation remain to be elucidated [71].

## 4. Perspectives

The functional role of PRDM2 in diseases is centered on the RIZ1 subtype. It not only serves as a potential diagnostic marker and therapeutic target in multiple systemic malignancies but also demonstrates extensive potential in inflammatory, metabolic, and neurodegenerative diseases.

Despite the significant progress made in recent years in the study of the role of PRDM2, especially the RIZ1 subtype, in multiple systemic diseases, there are still many scientific issues that need to be urgently addressed. Firstly, the current research on PRDM2 in the urinary and circulatory system diseases is still blank, and its specific functions in cardiovascular regulation, kidney development, and urinary tract tumors have not been systematically clarified, indicating a broad exploration space in these areas. Secondly, although RIZ1 is widely recognized as having tumor suppressor activity, the biological function of the RIZ2 subtype and its specific mechanism of action in tumorigenesis remain unclear, and the dynamic impact of the disruption of the RIZ1/RIZ2 balance on the disease process awaits in-depth study. Additionally, most current studies on PRDM2 focus on the epigenetic regulation mediated by DNA methylation, while the integrated regulatory mechanism of its involved protein interaction networks and upstream and downstream signal pathways still lacks systematic analysis, limiting the development of targeted intervention strategies.

Future research on PRDM2 should focus on the following key directions. First, integrating multi-omics approaches—including transcriptomics, proteomics, epigenomics, and metabolomics—will be essential to elucidate the dynamic regulatory patterns of PRDM2 under various physiological and pathological conditions. Second, the development of isoform-specific small-molecule agonists or inhibitors may offer a promising avenue for precise therapeutic intervention. Although RIZ1 is widely recognized as a tumor suppressor, emerging evidence suggests that certain isoforms or splicing variants—such as RIZ1v2 or RIZ2—may exert oncogenic functions in specific cellular contexts. Therefore, activating canonical RIZ1 while inhibiting its oncogenic counterparts could provide a balanced and targeted therapeutic strategy.

Additionally, constructing tissue-specific PRDM2 knockout models will help uncover its roles across different organ systems and developmental stages. Further mechanistic exploration of PRDM2 in immune modulation and inflammatory signaling may expand its relevance to autoimmune and infectious diseases. Finally, promoting the clinical translation of PRDM2 research—including evaluating its potential as a diagnostic and prognostic biomarker, and developing personalized therapeutic regimens based on PRDM2-related molecular pathways—could significantly enhance its impact in precision medicine.

## 5. Conclusions

PRDM2, as a type of epigenetic regulatory factor with a highly conserved structural and functional nature, shows significant biological significance and clinical application potential in multiple systemic diseases. Therefore, we believe that further in-depth exploration of its mechanism of action and targeted regulatory strategies can provide a more precise and efficient theoretical basis and practical approach for the precise treatment of various difficult diseases.

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
