# Peer review of "PRDM2—The Key Research Targets for the Development of Diseases in Various Systems"

_biomolecules, 2025, doi:10.3390/biom15081170_

Round 1
Reviewer 1 Report
Comments and Suggestions for Authors
The work on PRDM2 is valuable; it represents the complexity of the topics developed and demonstrates the strengths of the data collected for progress. Some small issues remain for improvement.
Figura 1: Interessante utilizzo del software AlphaFold2 per la previsione 3D della struttura delle proteine. Questi dati sembrano essere inediti, o sono solo mie impressioni. Si prega di fornire il sito web o, se ottenuto diversamente, e le condizioni di trattamento.
Sarebbe necessario citare l'articolo doi:10.1128/MCB.24.16.7032-7042.2004. Credo che spieghi la relazione tra la funzione di RIZ come coattivatore di RIZ1, il recettore degli estrogeni ormono-dipendente. RIZ1 è essenziale per la risposta d'organo ormono-dipendente perché è assente nei modelli murini che eliminano RIZ1 pur continuando ad esprimere RIZ2.
Author Response
Thank you for your valuable comment. Please refer to the appendix for details.

Reviewer 2 Report
Comments and Suggestions for Authors
The subject and premise of the manuscript are interesting and potentially useful for a wide range of readership. The authors collected and revised a sufficient number of publications to demonstrate the different roles and activities of PRDM2 isoforms. However, the structure and style of the manuscript needs improvement before publication. My main concern is that the current version little more than a collection of loosely related quotes from the literature, lacking a clear flow and offering little clarity in terms of scientific content.
My specific comments:
1. The nomenclature used in the manuscript needs to be clarified and gene/protein names need to be used consciously.
The first three paragraphs of the 2.1. section need to be re-written as they are confusing in the current version.
Line 71: PRDM2 (RIZ) is identified, but it is not specified, which isoform.
Line 73: a PRDM2 variant is identified - again, no indication of canonical isoform name.
Line: 89: Transcipt 3, also RIZ2 isofom c is introduced. However, later in the paragraph only RIZ1 and RIZ2 are mentioned, no isoform c is present. I suggest the authors clearly define which isoform is referred by which name and stick to these names throughout the whole text. I also suggest to include UniProt isoform IDs when introducing a protein variant.
Later in the text, PRDM2 is often used without specifying the isoform - given the different functions of RIZ1/RIZ2, it would be important to use clear identification. In case the original publication didn't specify the isoform, this should be unequivocally stated and the results discussed with that in mind.
2. The names of the domains are also used inconsistently: at one point PR-Set7 domain is mentioned, but it is not clarified if this is the same as the PR structural domain. I suggest to include a schematic figure with the different isoforms and domains represented with clear names for each and stick to those throughout the text.
3. Clear contradictions in protein function are listed in the text without addressing them and discussing the possible reasons. E.g. RIZ1 is a tumor-supressor in line 109 and an oncogene in line 113; in line 240 "The expression of RIZ1 is reduced in hepatocellular carcinoma tumors" and in line 252 "RIZ1 is reportedly upregulated in hepatocellular carcinoma cells". These require much more detailed discussion.
4. In many instances random information is given without any explanation as to why that is relevant in terms of function/disease. E.g. in lines 252-354, HOTAIRM1 and miR-125b are mentioned, but no details are offered how they are relevant.
5. In most of the examples, RIZ1 acts as tumor suppressor and its lower expression is related to cancer progression. However, the authors suggest that "...developing and applying specific small molecule agonists or inhibitors to target and regulate the activity of RIZ1 to achieve precise intervention in malignant tumors or metabolic diseases..." (lines 425-426) is a promising future research direction. I suggest to add more explanation as to why an inhibitor would be a good choice for targeting a tumor suppressor, or in which specific cases would inhibition be a promising strategy.
Minor comments:
- Lines 270-274 essentially repeat the same sentence.
- At numerous places sentences like this "These findings provide new insights into the tumor-suppressor mechanism of RIZ1." (line 328) feel like they originate from the cited publications and are not directly relevant in the current review without further elaboration. I suggest to remove or rephrase these.
Thorough English editing is necessary as the current version contains several hard-to-understand sentences and phrases.
Some examples:
- Line 41: "PRDMs are a subfamily of Kruppel-like zinc finger (ZF) gene proteins, first in post-natal/multicellular animals..." - it is unclear what the authors mean by "post-natal" animals.
- Line 75: "This variant is downregulated by binding to the cis-regulatory DNA element GTCATTGAC (MTE: macrophage-specific TPA-responsive element), which is responsible for the induction of the human heme oxygenase-1 gene during macrophage differentiation." - it is unclear how the downregulation happens.
- Line 107: "mRNA expression deletion" and "shifted-code mutations" - I assume the authors meant frame-shift mutations, but I'm uncertain as to what the first term relates to. Is it mRNA silencing maybe?
- Line 108: "It is also a tumor-susceptible gene in mice."
- Line 309: "In conclusion, inactivating the PRDM2/RIZ gene leads to promoter hypermethylation and thus represses the expression of the oncogenic isoform RIZ1. In turn, biological functions are altered and carcinogenesis is mediated in major organs of the digestive system."
- Line 350: "...RIZ1 is the only negative choice but not RIZ2 in breast cancer." - it is unclear what the authors meant by "negative choice".
Author Response
Thank you for your valuable comment. Please see the attachment.

Round 2
Reviewer 2 Report
Comments and Suggestions for Authors
The authors responded to my comments and the updated manuscript improved significantly in clarity.